

# A strategy to find novel candidate anti-Alzheimer's disease drugs by constructing interaction networks between drug targets and natural compounds in medical plants

Bi-Wen Chen[1,2,*], Wen-Xing Li[2,3,*], Guang-Hui Wang[1], Gong-Hua Li[2,3], Jia-Qian Liu[4], Jun-Juan Zheng[2,3], Qian Wang[2,3], Hui-Juan Li[2,3], Shao-Xing Dai[2,3] and Jing-Fei Huang[2,3,5]

[1] College of Pharmaceutical Sciences, Soochow University, Suzhou, Jiangsu, China
[2] State Key Laboratory of Genetic Resources and Evolution, Kunming Institute of Zoology, Chinese Academy of Sciences, Kunming, Yunnan, China
[3] Kunming College of Life Science, University of Chinese Academy of Sciences, Kunming, Yunnan, China
[4] School of Life Sciences, Zhengzhou University, Zhengzhou, Henan, China
[5] KIZ-SU Joint Laboratory of Animal Models and Drug Development, College of Pharmaceutical Sciences, Soochow University, Kunming, Yunnan, China
[*] These authors contributed equally to this work.

Corresponding authors
Shao-Xing Dai,
daishaoxing@mail.kiz.ac.cn
Jing-Fei Huang,
huangjf@mail.kiz.ac.cn

## ABSTRACT

**Background**. Alzheimer' disease (AD) is an ultimately fatal degenerative brain disorder that has an increasingly large burden on health and social care systems. There are only five drugs for AD on the market, and no new effective medicines have been discovered for many years. Chinese medicinal plants have been used to treat diseases for thousands of years, and screening herbal remedies is a way to develop new drugs.

**Methods**. We used molecular docking to screen 30,438 compounds from Traditional Chinese Medicine (TCM) against a comprehensive list of AD target proteins. TCM compounds in the top 0.5% of binding affinity scores for each target protein were selected as our research objects. Structural similarities between existing drugs from DrugBank database and selected TCM compounds as well as the druggability of our candidate compounds were studied. Finally, we searched the CNKI database to obtain studies on anti-AD Chinese plants from 2007 to 2017, and only clinical studies were included.

**Results**. A total of 1,476 compounds (top 0.5%) were selected as drug candidates. Most of these compounds are abundantly found in plants used for treating AD in China, especially the plants from two genera Panax and Morus. We classified the compounds by single target and multiple targets and analyzed the interactions between target proteins and compounds. Analysis of structural similarity revealed that 17 candidate anti-AD compounds were structurally identical to 14 existing approved drugs. Most of them have been reported to have a positive effect in AD. After filtering for compound druggability, we identified 11 anti-AD compounds with favorable properties, seven of which are found in anti-AD Chinese plants. Of 11 anti-AD compounds, four compounds 5,862, 5,863, 5,868, 5,869 have anti-inflammatory activity. The compound 28,814 mainly has

immunoregulatory activity. The other six compounds have not yet been reported for any biology activity at present.

**Discussion**. Natural compounds from TCM provide a broad prospect for the screening of anti-AD drugs. In this work, we established networks to systematically study the connections among natural compounds, approved drugs, TCM plants and AD target proteins with the goal of identifying promising drug candidates. We hope that our study will facilitate in-depth research for the treatment of AD in Chinese medicine.

## INTRODUCTION

Alzheimer's disease (AD), a progressive and ultimately fatal degenerative brain disorder, is thought to be one of the main causes of dementia in senior citizens (*Fan & Chiu, 2014*; *Song et al., 2015*) (*Fan & Chiu, 2014*; *Song et al., 2015*). The psychiatric symptoms observed in AD patients, include irritability, changes in mood or personality, paranoid delusions and hallucinations (*Coyle, Price & DeLong, 1983*). The pathological features of AD include senile plaques and neurofibrillary degeneration (*Dickson, 1997*). Degeneration, caused by neurofibrillary tangles (intracellular fibrous aggregations of tau protein), mainly occurs in brain regions involved in learning, memory, and emotional behaviors, such as the hippocampus, basal forebrain, entorhinal cortex and amygdala (*Mattson, 2004*). The various hypotheses regarding AD pathogenesis suggest the involvement of many pathways and target proteins, such as the amyloid (*Goedert & Spillantini, 2006*), cholinergic (*Craig, Hong & McDonald, 2011*), oxidative stress (*Pratico, 2008*), glutamatergic (*Bezprozvanny & Mattson, 2008*), inflammatory (*Trepanier & Milgram, 2010*) and metal hypotheses (*Bonda et al., 2011*). However, the causes of AD remain unclear due to the complexity of this multifactorial disease (*Armstrong, 2013*). To date, five symptom-relieving drugs are available to AD patients in a clinical setting, including four cholinesterase inhibitors and one N-methyl-D-aspartate (NMDA) receptor antagonist. However, there is currently no method for reversing or curing AD (*Cummings, Morstorf & Zhong, 2014*; *Peng et al., 2016*), and Tacrine has been discontinued in the United States market. Thus, the discovery of new drugs for treating AD patients remains a challenge.

Traditional Chinese Medicines (TCMs) have been used in therapy and for treating various diseases for several thousand years of Chinese history, and some natural ingredients in TCMs have been successfully developed into drug, such as artemisinin. Screening natural ingredients or compounds from herbal remedies and TCMs may be an effective way to develop new drugs (*Normile, 2003*; *Sanderson, 2011*; *Sucher, 2013*). For example, interactions between some ingredients from anti-AD herbs and corresponding anti-AD target proteins (*Sun et al., 2013*) as well as between 12 ginger components and 13 anti-AD target proteins have been reported (*Azam et al., 2014*). Many AD target proteins have previously been validated, including AchE (*Yiannopoulou & Papageorgiou, 2013*), BchE (*Darvesh, 2016*; *Mushtaq et al., 2014*), RAGE (*Cai et al., 2016*; *Deane, 2012*), TNF-alpha

((*Leszek et al., 2016*); (*Wyss-Coray & Rogers, 2012*), PLA2 (*Gentile et al., 2012*; *Lee et al., 2011*) and others. These proteins are involved in a variety of AD-associated pathways. Because we wanted to study a comprehensive range of AD target proteins, we selected 30 target proteins with protein crystal structures (including ligand present in the crystal structure) from all of the validated AD therapeutic target proteins provided by Thomson Reuters Integrity database as our research objects. To explore the interactions between the 30 validated AD therapeutic target proteins, which represent a variety of hypotheses regarding AD pathogenesis, and compounds in TCM plants, we established interaction networks among the target proteins, compounds, approved drugs and TCMs. Finally, we identified 11 structurally novel candidate anti-AD compounds with favorable druggability properties, seven of which are found in anti-AD Chinese plants. The 11 compounds identified in this study may be valuable in future anti-AD drug development, though further experiments are needed to prove their usefulness as drugs. The results suggest that the strategy of interaction network-based drug discovery may be very helpful for drug development.

## MATERIALS AND METHODS

### Data collection and preprocessing

More than 60,000 natural compounds from 8,529 different plants were from the TCM Database@Taiwan (http://tcm.cmu.edu.tw/). This web-based database is the most comprehensive non-commercial database of TCM(*Chen, 2011*). We obtained the 3D structures of the compounds from the database as mol2 files and converted them to the pdbqt format and SMILES string using Open Babel toolbox v2.3.1(*O'Boyle et al., 2011*).

The validated therapeutic AD target proteins and pathways were provided by Thomson Reuters Integrity database (https://integrity.thomson-pharma.com/integrity/). Studies on the target proteins were obtained using Pubmed and PMC (https://www.ncbi.nlm.nih.gov/pubmed). The target proteins structures were obtained from the Protein Data Bank (PDB) database (http://www.rcsb.org/pdb/home/). We selected target proteins of interest by determining whether studies relating them to AD and crystal protein structures (with ligand present) were available. Inhibitors or agonists were used to confirm the ligands in the references we identified. We ultimately selected 30 target proteins. Information regarding the 30 target proteins, such as the Integrity name, corresponding name in the Uniprot database (http://www.uniprot.org/), Uniprot ID, PDB ID, ligand ID and document IDs in Pubmed and PMC are available in Table S1. The 3D structures of the proteins are available as files in the pdb format. This format was converted to the pdbqt format using AutoDock tools v1.5.6 (*Morris et al., 2009*), and the 3D view was generated by Discovery Studio v3.1 (http://accelrys.com/products/collaborative-science/biovia-discovery-studio/).

### Molecular docking between natural compounds and AD target proteins

Docking is tantamount to position the ligand in different orientations and conformations within the binding site to calculate optimal binding geometries and energies. Interactions between natural compounds and the AD target proteins were previously predicted using

AutoDock Vina 1.1.2 (*Trott & Olson, 2010*). The docking binding site center for each target protein is the structural binding center of the ligand present in the crystal structure. The ligands were confirmed by the studies we identified. The coordinates of the docking center, ligand ID in PDB database and supporting documents are shown in Table S1. To allow free rotation of the compounds, the search space was set to $25 \times 25 \times 25$ Å in each axis. The default settings were used for all of the other docking parameters. Each docking was performed by a command that contained the space size and three-dimensional coordinate of the docking center. For each compound, the binding pose with the lowest energy for each docking test was considered the best binding mode. A lower energy score indicated a stronger binding affinity between the ligand and receptor. The compounds with the top 0.5% docking score were chosen as the candidate ligands for each target protein.

### Validation of the docking results

To validate the docking results, three methods were used. First, we manually checked the docking results and visualized the interaction between the compound and receptor to verify that the compound was in the binding pocket. Second, the original ligand in the crystal structure was set as the reference. The docking energy of candidate anti-AD compounds should be better than or comparable to that of original ligand. Third, we check whether the top 0.5% TCM compounds are similar to the existing drugs that have been reported in anti-AD research.

### Interactions among the target proteins, compounds and plants

The interaction between each target protein and its best-binding TCM compound was displayed using the PyMOL (PyMOL Molecular Graphics System, version 1.7) program (https://pymol.org). The pharmacophore was displayed using Discovery Studio v3.1.

The target-compound and target-plant networks were constructed using Cytoscape v3.4.0 (*Shannon et al., 2003*). In these networks, the target protein and compound were considered to be connected if the compound successfully docked to the target protein, and the target protein and the plant were considered to be connected if the plant with the compound was able to interact with the target protein. The strength of the links is represented by the line' thickness, which indicates the number of compounds shared between the target protein and plant.

### Collection of anti-AD plants from Chinese medicine prescription

The term ''senile dementia'' was searched in the subject column of the CNKI database (http://www.cnki.net/) to retrieve Chinese medicine prescriptions for anti-AD from the relevant Chinese articles. Articles from clinical studies between 2007 and 2017 were selected. Chinese medicine prescriptions and the usage frequency were also obtained from these articles. The common anti-AD plants in traditional Chinese clinical medicines were identified from the prescriptions. For reference, Table S2 shows information regarding the Chinese version of the raw prescription data with corresponding English, the Latin name of the anti-AD plants in each prescription, the patient number (male and female if available), the article title of the study, published data (years) and the article format.

## Similarity between candidate compounds and existing drugs

The Tanimoto coefficient (Tc) and Pybel (*O'Boyle, Morley & Hutchison, 2008*) Python package were used to measure the structural similarities between compounds. The fingerprint FP2 implemented in Pybel was generated for each structure and used to calculate Tc. Tc is defined as Tc = C(i, j)/U(i, j), where C(i, j) is the number of common features in the fingerprints of molecules i and j and U(i, j) is the number of all of the features in the union of the fingerprints of molecules i and j. If the fingerprints of two compounds are Tc = 1, even if they differ among themselves by isolated instances of C, N or O atoms, they will be considered structurally identical.

Cytoscape v3.4.0 was used to construct a network linking the candidate compounds, their target proteins and structurally identical drugs. A natural compound and an existing drug in the DrugBank database (*Wishart et al., 2006*) were considered to be connected if their Tc score was 1. The natural compounds and their target proteins were also connected in this network.

## Clusters of potential candidate compounds for AD

One-thousand-four-hundred-seventy-six compounds located in the top 0.5% of all compounds that docked with 30 target proteins were regarded as potential candidate compounds for AD. The cluster ligands protocol in BOVIA Pipeline Pilot V8.5 (http://accelrys.com/products/collaborative-science/biovia-pipeline-pilot/) was used to cluster the compounds. During clustering, a set of compounds was assigned to different clusters based on the similarity of their properties. Clustering was based on the root-mean-square (RMS) difference of the descriptor properties or the Tanimoto distance for fingerprints. In our study, we clustered the compounds based on the Tanimoto distance using the fingerprint FP2. Cluster ligands were performed by the number of size or the number of molecules per cluster. The default parameter in Pipeline Pilot V8.5 was set to fixed number of 10 clusters. For simplicity, we used default parameters to cluster our compounds. The molecule with the lowest total distance to all other members of the cluster was considered the cluster center.

## ADMET and logP properties for candidate compounds for AD

The ADMET and logP properties of candidate compounds for AD were estimated using Discovery Studio. ADMET refers to absorption, distribution, metabolism, excretion and toxicity and logP refers to the logarithm of the partition coefficient. The ADMET properties, including aqueous solubility, blood brain barrier penetration (BBB), human intestinal absorption (HIA), plasma protein binding (PPB) and hepatotoxicity, as well as logP were used to filter the compounds. The values of these properties were set as the controlled parameters, which were 3~4 (3: good; 4: optimal) for aqueous solubility, 1~2 (1: high; 2: medium) for BBB, 0 (0: good) for HIA, FALSE for both PPB and hepatotoxicity and logP <5.

## RESULTS

### Molecular docking of natural compounds and embedded ligands to the 30 AD target proteins

Of the 60,000 compounds in the TCM Database, 30,438 contain plant information, and these compounds were docked with the 30 selected AD target proteins. The original ligands of each target protein were also docked to their corresponding target proteins. The docking results are shown in Table 1. The docking scores of ligands embedded in the protein crystal structure ranged from −3.31 to −12.65 (kcal/mol). The lowest docking energy scores for the 30 target proteins ranged from −8.44 to −14.5 (kcal/mol). Some target proteins, such as Caspase-3, QC, IDO and GLP-1R, were able to bind over 20,000 natural compounds with docking scores superior to those of their embedded ligands. However, the docking scores of the target RAR with natural compounds were inferior to those of its embedded ligand.

Because many TCM compounds can bind to AD target proteins, for each target protein, only the top 0.5% of compounds in terms of docking scores (a total of 1,476 compounds) were selected as candidate compounds for AD. The original study of AutoDockVina showed that the success rate of Vina is 80% (RMSD < 2) for an independent validation dataset. Furthermore, Vina achieves a low standard error of 2.85 kcal/mol compared with the experimental free energies. There is a highly positive correlation between the predicted and experimental free energies of binding. In our study, almost all of the docking energies of the top 0.5% of compounds bound to target proteins were superior to those of their embedded ligands (Fig. 1). Thus, the 1,476 compounds are likely candidate compounds for AD.

### Analysis of the interactions between target proteins and ligands including TCM compounds, original ligands and approved AD drugs

The docking pose interactions between the target proteins and ligands including their best-binding TCM compounds and original ligands are shown in Fig. 2A and Figs. S1 to S29. The figures show side-by-side-comparisons of best TCM ligands and original ligands. The best TCM ligands and original ligands are located in the same binding pocket for each target protein and they all have some common residues. Taking AchE as an example: the best-binding TCM compound for AchE is 24,829, the original ligand is Huperzine A, and their common residues are TYR-124, PHE-297, PHE-338, TYR-337, ASP-74. Furthermore, we compared the structures of the best-binding TCM compounds and previously known ligands (Table S3) and found the structures to be different. The Tc scores are between 0.06 and 0.48. Therefore, most of the best-binding TCM compounds are novel scaffolds.

We analyzed the pharmacophore of SIRT1 using the top 10 TCM compounds binding the target protein (Fig. 2B). The pharmacophore model consists of one hydrogen bond acceptor (HBA, green) and five hydrophobic centers (blue); therefore, we think that compounds that have this model may easily bind with SIRT1.

We also compared the docking energy scores between the approved AD drugs and the candidate compounds for the protein AchE (Table S4). The approved AD drugs that we analyzed were Donepezil, Galantamine and Rivastigmine. All the three drugs were AchE

**Table 1   Details of the docking results of 30 anti-AD targets with the number of successfully docked TCM compounds.**

| RCSB ID | Protein name | Original ligands ID | Binding energy of original ligand[a] | Lowest docking energy | Compound number[b] |
|---|---|---|---|---|---|
| 1DB4 | PLA2(Phospholipase A2, membrane associated) | 8IN | −7.31 | −11.55 | 5,290 |
| 1DQA | HMG-COA(3-hydroxy-3-methylglutaryl-coenzyme A reductase) | NAP | −7.42 | −9.78 | 437 |
| 1NME | Caspase-3 | 159 | −4.57 | −10.24 | 21,028 |
| 1OJA | MAOB(Amine oxidase [flavin-containing] B) | ISN | −6.58 | −12.2 | 11,173 |
| 1TB7 | PDE4(cAMP-specific 3′,5′-cyclic phosphodiesterase 4D) | AMP | −6.47 | −14.5 | 17,375 |
| 1TN6 | Ftase(Protein farnesyltransferase subunit beta) | FII | −6.59 | −11.9 | 14,437 |
| 2AFW | QC(Glutaminyl-peptide cyclotransferase) | AHN | −4.48 | −11.11 | 23,635 |
| 2AZ5 | TNF(Tumor necrosis factor) | 307 | −5.66 | −9.53 | 9,261 |
| 2D0T | IDO(Indoleamine 2,3-dioxygenase 1) | PIM | −5.71 | −12.4 | 20,739 |
| 2DQ7 | Fyn(Tyrosine-protein kinase Fyn) | STU | −10.28 | −12.41 | 63 |
| 2VQM | HDAC(Histone deacetylase 4) | HA3 | −7.11 | −11.33 | 5,356 |
| 2Z5Y | MAOA(Amine oxidase [flavin-containing] A) | HRM | −7.96 | −12.8 | 5,299 |
| 3A4O | lyn(Tyrosine-protein kinase Lyn) | STU | −9.4 | −12.53 | 431 |
| 3G9N | JNK(Mitogen-activated protein kinase 10) | J88 | −7.19 | −10.36 | 1,606 |
| 3IKA | EGFR(Epidermal growth factor receptor) | 0UN | −7.64 | −11.45 | 6,324 |
| 3KMR | RAR(Retinoic acid receptor alpha) | EQN | −12.65 | −11.4 | 0 |
| 3O3U | RAGE(Advanced glycosylation end product-specific receptor) | MLR | −7.76 | −14.08 | 13,309 |
| 4DJU | BACE-1(Beta-secretase 1) | 0KK | −7.12 | −12.2 | 14,161 |
| 4EY5 | AchE(Acetylcholinesterase) | HUP | −8.5 | −10.6 | 329 |
| 4MS4 | GABA(B)(Gamma-aminobutyric acid type B receptor subunit 1) | 2C0 | −5.73 | −10.6 | 13,107 |
| 4OC7 | RXR(Retinoic acid receptor RXR-alpha) | 2QO | −8.48 | −11.3 | 708 |
| 4XAR | MGLUR(Metabotropic glutamate receptor 3) | 40F | −4.98 | −8.5 | 9,244 |
| 4YLK | DYRK1A(Dual specificity tyrosine-phosphorylation-regulated kinase 1A) | 4E2 | −8.13 | −12.54 | 7,167 |
| 4ZGM | GLP-1R(Glucagon-like peptide 1 receptor) | 32M | −3.31 | −9.06 | 24,782 |
| 4ZZJ | SIRT1(NAD-dependent protein deacetylase sirtuin-1) | 4TQ | −6.89 | −8.86 | 108 |
| 5A46 | FGFR1(Fibroblast growth factor receptor 1) | 38O | −8.54 | −12.8 | 699 |
| 5AFH | α7NACHR(Neuronal acetylcholine receptor subunit alpha-7) | L0B | −6.02 | −9.64 | 6,934 |
| 5H8S | AMPA(Glutamate receptor 2) | 5YC | −5.3 | −8.44 | 8,926 |
| 5HK1 | SIG-1R(Sigma non-opioid intracellular receptor 1) | 61W | −9.29 | −12.8 | 1,281 |
| 5IH5 | CKI-δ(Casein kinase I isoform delta) | AUE | −7.62 | −12.5 | 5,998 |

**Notes.**
[a]'Binding Energy of Original Ligand' indicates the docking energy of the ligand embedded in the crystal structure.
[b]The number of compounds with better docking scores than that of the original ligand embedded in the crystal structure.

inhibitors. The average docking energy score of the candidate compounds was superior to that of the approved AD drugs for the protein AchE. The docking pose interaction of the three drugs and the best-binding TCM compound with the protein AchE is shown in Fig. S30. The three drugs and best-binding TCM compound were located in the same binding pocket, and their docking poses were different because their structures were different.
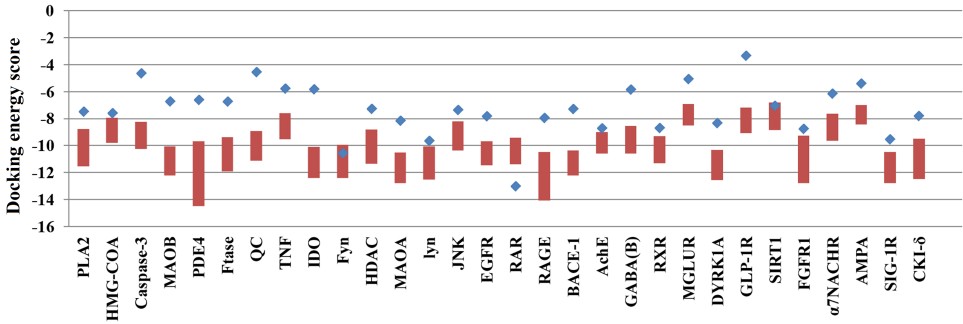

**Figure 1** **The docking energy scores of the top 0.5% TCM compounds and original ligands for 30 targets.** Red boxes represent the top 0.5% of compounds for each target protein. Blue points represent the target proteins' embedded ligands.

## Analysis of single-target and multi-target compounds

Among the 1,476 candidate compounds for AD, there were 850 compounds with a single target and 626 compounds with multiple targets (see Figs. S31 and S32). The single-target compounds corresponding to each target varied widely in number. For example, in single-target networks, target SIRT1 corresponds to 30 compounds, whereas target lyn just corresponds to four compounds. The multi-target compounds were classified into 18 networks based on their corresponding target numbers, which ranged from 2 to 24. As the number of targets per compound increased, the number of compounds in that category decreased. For example, the two-target network contained 260 compounds, whereas the three-target and four-target networks contained 90 and 77 compounds, respectively. Finally, we observed that compound 24,508 could bind to 25 AD target proteins. The structure and network of compound 24,508 are shown in Fig. S32.

## Candidate AD compounds and their enrichment plants

We mapped 1476 candidate AD compounds (corresponding to 30 AD target proteins) to the 334 plants. The plant numbers for each target protein ranged from 42 to 71, whereas the compound numbers for each target protein ranged from 62 to 132 (Fig. 3 and Fig. S33).

We selected 101 clinically related studies out of over 10,000 senile dementia-related articles and identified 141 anti-AD traditional Chinese plants from the clinical prescriptions in these articles. The 141 traditional Chinese anti-AD plants were classified based on their functional properties in the TCM database. Most of the 141 anti-AD plants were in the 'Tonifying, Replenishing' category, and the plants in this category accounted for 28.45% of all anti-AD plants (Fig. S34).

The best-associated plant for each target protein contained the greatest number of compounds capable of docking with the target protein (Table 2 and Fig. 4). The number of compounds in Table 2 was based on the top 0.5% of compounds successfully docking to each target protein. Thus, 30 target proteins corresponded to 16 best-associated plants, of which the top five plants were anti-AD traditional Chinese plants, including *Panax* and *Morus*, which corresponded to 7 and 5 target proteins, respectively, as well as *Salvia, Rheum* and *Paeonia*.

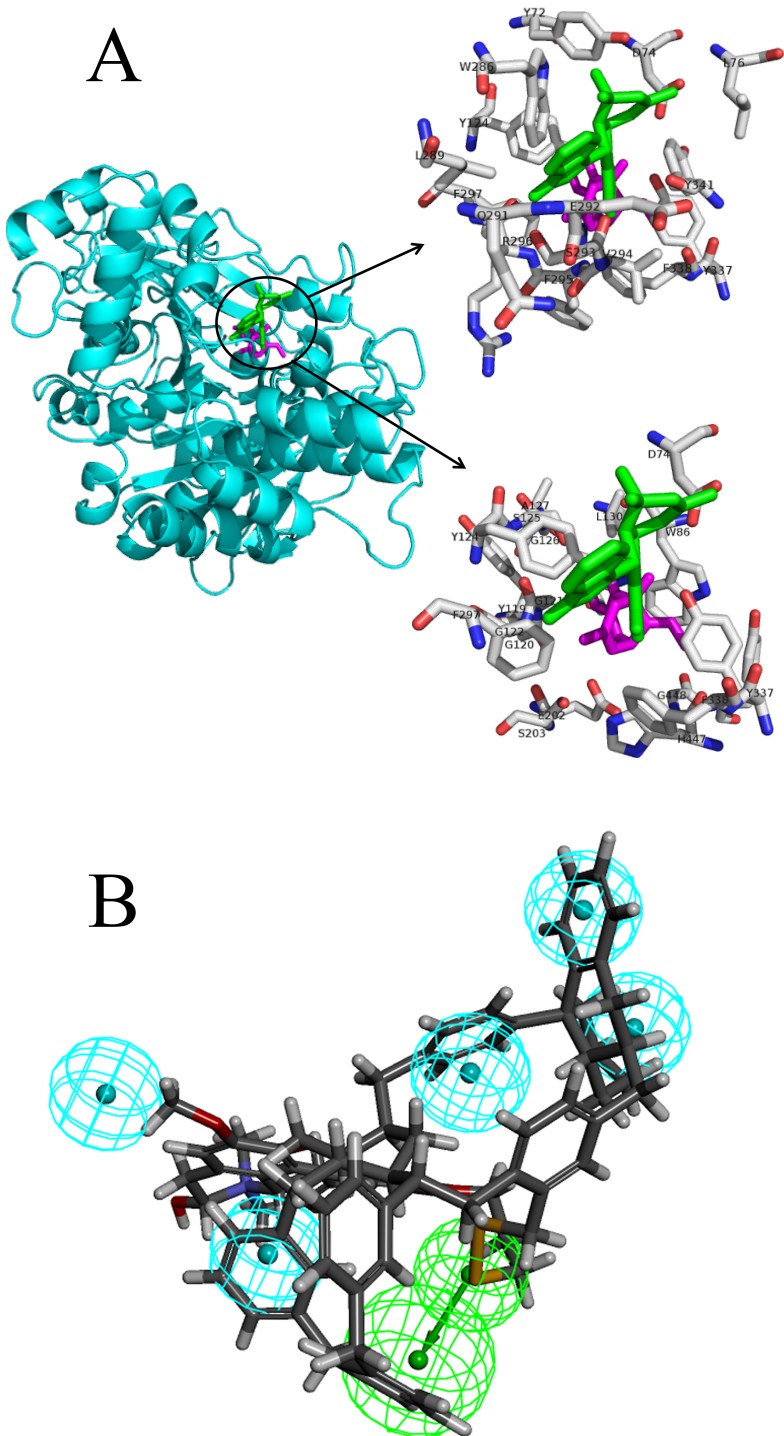

**Figure 2   The docking pose interactions between the target proteins and ligands including their best-binding TCM compounds and original ligands and the pharmacophore of SIRT1.** (A) The 3D structures and binding model of ligands including best ligand and original ligand to the target protein AchE. The best ligand is green and the original ligand is magenta. The top panel shows the amino acid residues lying within 5 Å from the best ligand, and the bottom panel shows the amino acid residues lying within 5 Å from the original ligand. (B) The pharmacophore of SIRT1 using top 10 TCM compounds binding for the target protein. The hydrogen bond acceptor is in green and the hydrophobic centers are in blue.

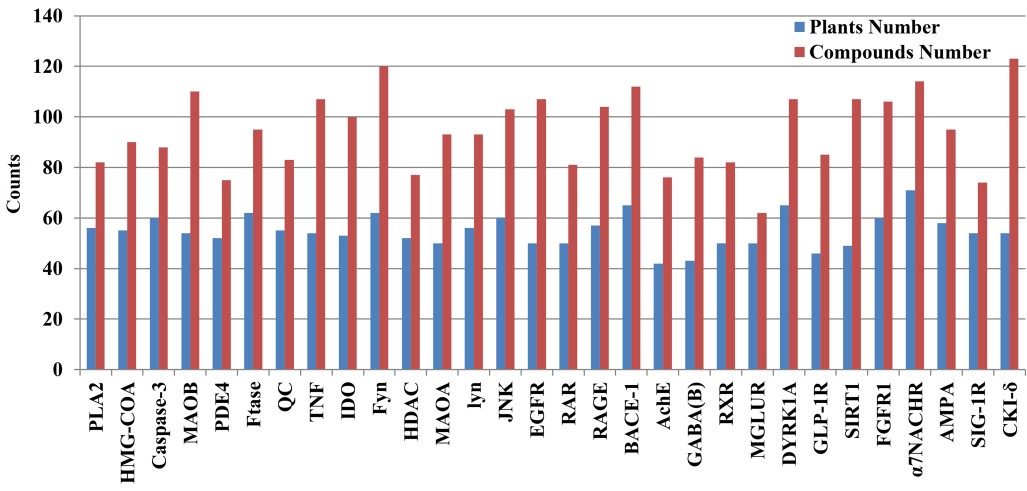

**Figure 3  The exact number of candidate anti-AD compounds and their plants associated with each anti-AD target protein.** The number is tagged above each column, and the target proteins are displayed on the horizontal axis.

**Table 2  AD targets and their best-associated plant with the most compounds docking with the target.**

| Target | Top1 Plant | Target | Top1 Plant | Target | Top1 Plant |
|---|---|---|---|---|---|
| PLA2 | Bletilla(5)[a] | HMG-COA | Morus(9) | Caspase-3 | Paeonia(4) |
| MAOB | Corydalis(16) | PDE4 | Isatis(4) | Ftase | Panax(8) |
| QC | Panax(4) | TNF | Panax(10) | IDO | Morus(7) |
| Fyn | Papaver(11) | HDAC | Bletilla(5) | MAOA | Corydalis(11) |
| lyn | Claviceps(5) | JNK | Morus(8) | EGFR | Artemisia(7) |
| RAR[b] | Rauwolfia(8) | RAGE | Fritillaria(7) | BACE-1 | Lonicera(6) |
| AchE | Piper(6) | SIRT1 | Panax(18) | GABA(B) | Morus(11) |
| RXR | Salvia(10) | MGLUR | Morus(4) | DYRK1A | Strychnos(6) |
| GLP-1R | Panax(9) | FGFR1 | Rheum(6) | α7NACHR | Panax(8) |
| AMPA | Panax(7) | SIG-1R | Corydalis(7) | CKI-δ | Salvia(11) |

**Notes.**
[a]The numbers in this table are compound numbers which the best-associated plant for each target protein contains.
[b]The docking energy of TCM compounds is higher than that of the original ligand for RAR protein.

## Similarities between candidate compounds and existing drugs

A structural comparison between all existing approved drugs recorded in the DrugBank and the top 0.5% of the natural compounds tested demonstrated that some compounds were identical to existing drugs ($Tc = 1$). The connection network among candidate compounds, existing drugs and AD target proteins was established, and the chemical structures of the compounds are also shown in Fig. 5. There were 17 candidate compounds, 14 existing drugs and 25 AD-associated target proteins in the network. The 14 drugs included Lutein (DB00137), Vitamin A (DB00162), Vitamin E (DB00163), Azelaic Acid (DB00548), Ergotamine (DB00696), Estradiol (DB00783), Menthol (DB00825),

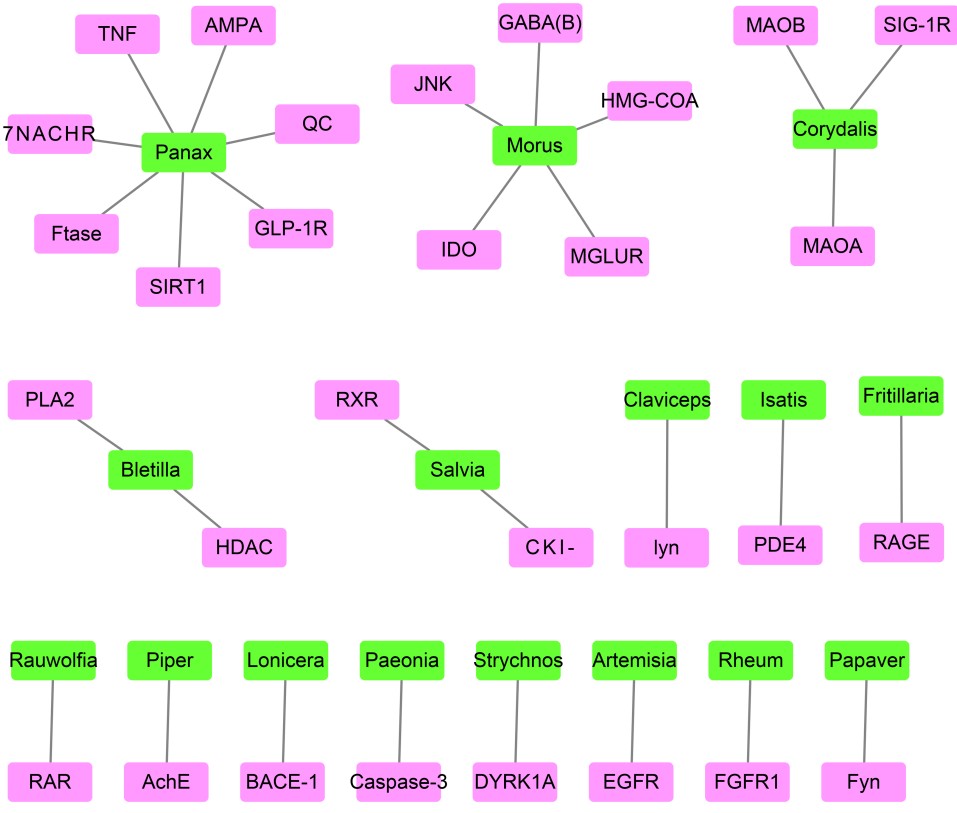

**Figure 4** **The network containing the target proteins and their best-associated plants.** Pink boxes represent target proteins. Green boxes represent compounds.

Drostanolone (DB00858), Glyburide (DB01016), Tubocurarine (DB01199), Metocurine (DB01336), Yohimbine (DB01392), Lactose (DB04465), Artemether (DB06697).

Most of the drugs listed above have been reported to have a positive effect in AD. Some studies have shown that Lutein is involved in preventing cognitive decline and decreasing the risk of AD; thus, Lutein may contribute to the treatment of AD (*Kiko et al., 2012*; *Min & Min, 2014*; *Xu & Lin, 2015*). Similarly, Vitamin A, Vitamin E, Estradiol, Menthol, Glyburide and Yohimbine are also considered useful in the prevention and treatment of AD (*Bhadania et al., 2012*; *Dysken et al., 2014*; *Lamkanfi et al., 2009*; *Lan et al., 2016*; *Mohamd et al., 2011*; *Ono & Yamada, 2012*; *Peskind et al., 1995*; *Takasaki et al., 2011*). Therefore, compounds with structures similar to the existing drugs may also have anti-AD function by interacting with similar target proteins.

Of the 17 candidate compounds, 11 can only bind with one target protein, whereas the rest, which are similar to the 14 drugs discussed above, can interact with more than one target protein. For example, compound 18491 which is similar to Menthol can only interact with the target Ftase and compounds 19,476 and 19,477 are similar to Tubocurarine and Metocurine, respectively, whereas compound 18,582 which is similar to Ergotamine can bind with 17 target proteins.

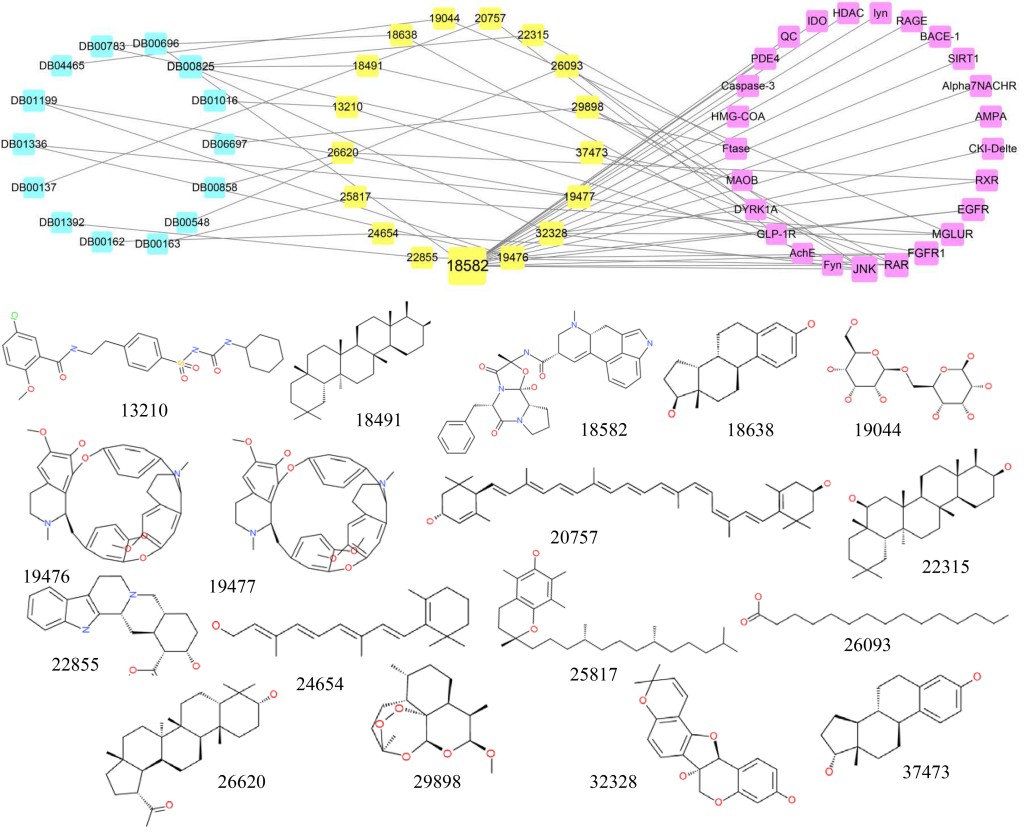

**Figure 5** **The network containing the anti-AD target proteins, TCM compounds and structurally identical drugs.** Pink boxes represent targets. Yellow boxes represent compounds. Blue boxes represent drugs. The structures of TCM compounds are also shown.

## Structure clusters of candidate AD compounds

To compare the structural features of the candidate AD compounds, the 1,476 candidate compounds were assigned to 10 clusters. The structures of the cluster center compounds and maximal common substructure of each cluster are shown in Fig. 6. The Generate Maximal Common Substructure component must contain the proportion of the cluster molecules. The proportion was set to 0.5 to find the largest maximal common substructure that was contained in at least 50% of the cluster molecules. All of the compounds in the center of the cluster contained the carbocyclic structure, similar to the five approved drugs for AD. The cluster sizes varied, with the largest containing 464 compounds and the smallest containing only six compounds. Every cluster had a primary target protein that could better combine with the compounds in the cluster. The binding of the best member of the cluster to the primary target protein and their structures are also shown in Fig. 6.

To see if there were similar or different scaffolds able to bind each target protein, clusters in the sets of ligands that bind to each individual protein were shown in Tables S5 to S35. Each set of ligands was assigned to 10 clusters. The compounds in the center of the cluster do not have the same scaffolds for each individual protein.

| Cluster | Cluster Center Structure | Maximal Common Substructure[a] | Best Member Structure | Cluster Size | Primary Targets |
|---|---|---|---|---|---|
| 1 | 24720 | | 20387 | 225 | SIG-1R (33) |
| 2 | 5625 | | 5631 | 6 | JNK(4), DYRK1A (4), CKI-δ(4) |
| 3 | 26945 | | 38891 | 277 | JNK (35) |
| 4 | 6001 | | 20111 | 63 | RAR (17) |
| 5 | 23801 | | 22299 | 57 | CKI-δ(16) |
| 6 | 10574 | | 8548 | 130 | FGFR1 (38) |
| 7 | 6477 | | 6518 | 140 | SIRT1 (50) |
| 8 | 35138 | | 22842 | 464 | Fyn (55) |
| 9 | 20757 | | 10684 | 73 | QC (12), HDAC (12), RAGE (12) |
| 10 | 8219 | | 8201 | 41 | SIRT1 (26) |

**Figure 6** **The 10 clusters of anti-AD TCM compounds and their primary targets.** The Generate Maximal Common Substructure component must contain the proportion of the cluster molecules. The proportion was set to 0.5 to find the largest maximal common substructure contained in at least 50% of the cluster molecules.

**Table 3  ADMET and logP properties of 11 candidate drugs.**

| Name(ID) | Solubility Level | BBB Level | Hepatotoxic Prediction | Absorption Level | PPB Prediction | logP | Targets |
|---|---|---|---|---|---|---|---|
| (3S)-1-(3,4-Dihydroxyphenyl)-7-(4-hydroxyphenyl)heptan-3-ol(5862) | 3 | 2 | FALSE | 0 | FALSE | 4.578 | AchE |
| (3S)-1-(3,4-Dihydroxyphenyl)-7-(4-hydroxyphenyl)-(6E)-6-hepten-3-ol(5863) | 3 | 2 | FALSE | 0 | FALSE | 4.134 | AchE,GABA(B), MGLUR |
| (3R)-1-(3,4-Dihydroxyphenyl)-7-(4-hydroxyphenyl)heptan-3-ol(5868) | 3 | 2 | FALSE | 0 | FALSE | 4.578 | GABA(B) |
| (3R)-1-(3,4-Dihydroxyphenyl)-7-(4-hydroxyphenyl)-(6E)-6-hepten-3-ol(5869) | 3 | 2 | FALSE | 0 | FALSE | 4.134 | AchE |
| pallidine(9593) | 3 | 2 | FALSE | 0 | FALSE | 1.913 | MAOB |
| 4,5-di-o-caffeoyl,quinic,acid(10639) | 3 | 2 | FALSE | 0 | FALSE | 3.477 | PDE4 |
| Anagyrine(16167) | 3 | 1 | FALSE | 0 | FALSE | 2.053 | AchE |
| Blestrin D(26629) | 3 | 2 | FALSE | 0 | FALSE | 4.578 | PLA2,QC,HDAC, JNK,GABA(B) |
| Dibothrioclinin II(28468) | 4 | 2 | FALSE | 0 | FALSE | 1.222 | Ftase,QC,HDAC, GLP-1R,AMPA |
| 5,7-Dihydroxy-6,8-dimethyl-3-(4′-hydroxy-3′-methoxybenzyl)chroman-4-one(28814) | 3 | 2 | FALSE | 0 | FALSE | 1.144 | RAR |
| Glabroisoflavanone A(30713) | 3 | 2 | FALSE | 0 | FALSE | 1.913 | MAOA |

## Eleven candidate compounds for AD with favorable ADMET and logP properties

Most orally approved drugs have favorable druggability properties. After ADMET and logP filtering, 11 compounds from 1,476 candidate anti-AD compounds possessed favorable properties (Table 3). The 11 compounds all have good HIA and don't have PPB and hepatotoxicity. Compound 28,468 has optimal aqueous solubility, and compound 16,167 has high BBB. Of the 11 compounds, eight were single-target compounds, and the remaining three were multi-target compounds. For example, compounds 5,868, 9,593, 10,639, 28,814 and 30,713 could only bind with one target protein (GABA(B), MAOB, PDE4, RAR and MAOA, respectively), whereas compounds 5,862, 5,869 and 16,167 shared one common target (AchE). Compound 5,863 was able to bind with three target proteins (AchE, GABA(B) and MGLUR), and compound 26,629 and 28,468 were able to interact with five target proteins. These 11 compound structures and their corresponding plants are shown in Fig. 7. The plants corresponding to seven of these compounds are regarded as anti-AD plants in TCM, including *Curcuma kwangsiensis, Poria cocos, Lindera aggregate, Ophiopogon japonicus (L. f.) Ker-Gawl.* and *Glycyrrhiza glabra.* Eleven compounds belong to different organic compound classes. Compounds 5,862, 5,863, 5,868, 5,869 belong to linear diarylheptanoids. Compound 10,639, 16,167, 26,629, 28,468, 28,814, 30,713 belongs to guanidines, pyridines, hydrophenanthrenes, angular pyranocoumarins, naphthyridines, 8-prenylated isoflavanones, respectively. We also checked the 11 comopund structures and existing research in pubchem database. Compounds 5,862, 5,863, 5,868, 5,869 have

| Compound ID | Structure | Plant | Compound ID | Structure | Plant |
|---|---|---|---|---|---|
| 5862 | | Curcuma kwangsiensis | 5863 | | Curcuma kwangsiensis |
| 5868 | | Curcuma kwangsiensis | 5869 | | Curcuma kwangsiensis |
| 9593 | | Lindera aggregate | 10639 | | Taraxacum mongolicum |
| 16167 | | Thermopsis lanceolata R. Br., Laburnum anagyroides, Sophora flavescens Alt., Sophora tonkinensis | 26629 | | Bletilla striata |
| 28468 | | Gerbera piloselloides Cass. | 28814 | | Ophiopogon japonicus (L. f.) Ker-Gawl. |
| 30713 | | Glycyrrhiza glabra | | | |

**Figure 7 2D structure and corresponding plants of 11 compounds with favorable ADMET properties.**

anti-inflammatory activity and compound 28,814 mainly has immunoregulatory activity. The other 6 compounds have not yet been reported for any biology activity at present.

## DISCUSSION

Candidate compounds from traditional Chinese plants provide a broad prospect for screening anti-AD drugs. We established a network between compounds in traditional Chinese plants and a comprehensive list of anti-AD target proteins involved in various

hypotheses. This network, which links compounds, TCM plants and target proteins, may be very helpful for anti-AD drug design.

During our manual assessment of the binding pockets and modes of compounds, we discovered that some receptor had a large binding pocket and some had their binding pocket exposed on the surface. Therefore, the number of successfully docked compounds for these receptors is more than that of other receptors. This result suggests that for these receptors, there may be many false positive compounds that do not bind to the receptor. Thus, we choose TCM compounds with top 0.5% docking scores as our objects to avoid false positive compounds that do not bind to the receptor.

ADMET is an important index in drug development. After filtering the compounds according to five ADMET properties, 11 candidate anti-AD compounds with novel structures remained. Among the 11 compounds, eight were single-target compounds and the remaining three had more than two target proteins. The structures and target proteins of these compounds are known, so they can be easily studied in future drug development, because these compounds have favorable druggability properties, they may become the promising candidate drugs for AD. Of course, further experiments are necessary to establish their viability as real candidate drugs.

Many compounds that bind to AD-associated target proteins were observed in plants that have not been used to treat AD in traditional Chinese clinical prescription. Thus, some previously non-anti-AD plants may become the anti-AD plants, which will provide more natural compound resources for AD drug discovery and be will also be beneficial for the development of TCMs.

## CONCLUSION

In summary, this study offers one strategy to find novel candidate anti-AD drugs from traditional Chinese plants by constructing interaction networks between AD target proteins and natural compounds in TCM plants. We got a total of 1,476 drug candidates (top 0.5% docked compounds for each target) using this strategy. Of 1,476 drug candidates, 17 candidate anti-AD compounds were structurally identical to 14 existing approved drugs. In addition, 11 anti-AD candidate compounds with favorable ADMET and logP properties were identified. Of 11 identified compounds, four compounds have anti-inflammatory activity, including compounds 5,862, 5,863, 5,868, 5,869 and compound 28,814 mainly has immunoregulatory activity. Other six compounds have not bioassay research in pubchem database at present. Further experiments are needed to verify our drug candidates. This strategy and identified drug candidates may be helpful for anti-AD drug discovery.

## ACKNOWLEDGEMENTS

We thank our colleagues, Drs. Yi-Cheng Guo and Shun-Mei Chen for helpful comments on the manuscript. We also thank four reviewers for valuable comments.

### Funding

This work was supported by the National Basic Research Program of China (Grant No. 2013CB835100), and the National Natural Science Foundation of China (Grant No. 31401142 and No. 31401137). The funders had no role in study design, data collection and analysis, decision to publish, or preparation of the manuscript.

### Grant Disclosures

The following grant information was disclosed by the authors:
National Basic Research Program of China: 2013CB835100.
National Natural Science Foundation of China: 31401142, 31401137.

### Competing Interests

The authors declare there are no competing interests.

### Author Contributions

- Bi-Wen Chen conceived and designed the experiments, performed the experiments, analyzed the data, contributed reagents/materials/analysis tools, prepared figures and/or tables, authored or reviewed drafts of the paper, approved the final draft.
- Wen-Xing Li performed the experiments, analyzed the data, prepared figures and/or tables, authored or reviewed drafts of the paper, approved the final draft.
- Guang-Hui Wang contributed reagents/materials/analysis tools, approved the final draft.
- Gong-Hua Li, Jia-Qian Liu, Jun-Juan Zheng, Qian Wang and Hui-Juan Li authored or reviewed drafts of the paper, approved the final draft.
- Shao-Xing Dai conceived and designed the experiments, performed the experiments, analyzed the data, authored or reviewed drafts of the paper, approved the final draft.
- Jing-Fei Huang conceived and designed the experiments, contributed reagents/materials/analysis tools, authored or reviewed drafts of the paper, approved the final draft.

### Data Availability

The raw data are provided in the Supplemental Files.

### Supplemental Information

Supplemental information for this article can be found online at http://dx.doi.org/10.7717/peerj.4756#supplemental-information.

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
