# Peer review of "A strategy to find novel candidate anti-Alzheimer’s disease drugs by constructing interaction networks between drug targets and natural compounds in medical plants"

_PeerJ, doi:10.7717/peerj.4756_

## Round 0.1 · original submission · Major Revisions

All the reviewers recommend revisions, and heavy revision. Please carry out the corrections as suggested by the reviewers, then check again English language. The supplementary materials also have to be updated. Please also see the attached PDF file with further remarks. The topic is very important and all the findings should be reproducible. I suggest not describe large number of AD target focusing on 1 or 2 examples only in more details.

·

Basic reporting

This manuscript describes the molecular docking of more than 30000 molecules (present in plants used in traditional chinese medicine) to 33 proteins proposed to be involved in the biogenesis of Alzheimer disease. This impressive amount of docking work is, however, not presented in the most favorable way nor analysed to the fullest:

A) the quality of the language in this manuscript is extremely poor. The manuscript must be checked very thoroughly by a professionally-proficient English speaker.

B) Precise information of the docking poses/putative interactions between ligand and proteins is completely absent, and this strongly limits my enjoyment of this paper. A description of the molecular features of the best ligands must be provided, at least highlighting whether they are relatively strightforward analogs of previously-known ligands or whether they are novel scaffolds (and in this case, describing the interactions common among members of the same cluster)

C) Figure 4 (and all figures in the Supporting Information) is very hard to read and not very informative due to the large number of labels, small font and lack of legends. I understand that networks are intrinsically hard to display, but I cannot help thinking that figure 4 would be much more informative if it were divided into different figures and (especially) included the chemical structures of the compounds.

Experimental design

line 105 "The docking binding site center for each target is the structural binding center of ligand embedded." I think that authors mean:

"The docking binding site center for each target is the structural binding center of the ligand present in the crystal structure." If that is so, authors should confirm that they used only structures of complexes of putative AD-targets with their natural ligands (or confirmed inhibitors). Otherwise, there is a possibility that authors have mistakenly used, for example, the structure of a protein with a non-specifically-bound molecule as initial guess for the correct binding site.

line 134: authors should clarify that using fingerprint FP2 a Tc=1 can be obtained from different molecules even if they differ among themselves by isolated instances of C, N or O atoms.


line 146:was the clustering based on the computed Tanimoto coefficients? Please clarify.

line 148: how did the authors select the number of clusters? Was it based on, e.g. , maximum distance allowed between members of a cluster, minimum distance between members of different clusters, etc?

line 148 "If the sum of one member distancing to other members reaches the minimum value, this member is selected as the cluster center." I think the authors mean: "The molecule with the lowest total distance to all other members of the cluster was taken as the cluster center"

Validity of the findings

Table 1: Please include the identity of the ligand present in the crystal structure as a new column before the "Ligand Energy". I think the table would benefit from relebeling the "Ligand Energy" column to "Binding energy of original ligand" and by moving the "Compound number" column to the extreme right, to ensure that readers do not mistakenly read the "Ligand energy" column as referring to the docked compounds.


line 163: the massive number of compunds (20000 in 30438) binding strongly to caspase-3, QC, IDO and GLP-1R is surprising and (if correct) seems to imply that those enzymes would be readily blocked in vivo by 2/3 of the secondary metabolites found in plants. This does not seem very plausible, since enzyme variants which are easily inhibited would tend to be deleterious to survival and therefore selected out of the population. Did the authors confirm that those compounds do block the active site, i.e. if their docking poitions superpose with those to the original ligand instead of binding to the protein surface without interfering with the active site?

line 168: typo I think the authors mean "0.5%" instead of "5%"

line 174 and following: authors often use the word "target" when they mean "target protein". This becomes confusing. The description of the number of compounds in two-target and three-target networks is not very informative. I would have preferred to have found , for example, the structure of the compound which binds 25 target proteins, an analysis of the chemical diversity of the 76 compounds which bind SIRT1, identification of pharmacophores, etc.

Table 2: The plant best associated with RAR is included, in spite of the poor binding of all compunds to RAR (compared to the original ligand).

Table 3 could be made more informative by highlighting, in each structure of the molecule at the cluster center, the precise core shared by all members of that cluster.The binding energy of the molecule at the cluster center should also be compared to the binding of the best member of the cluster, to help identify key portions of the molecule

Additional comments

Replace "AD" by "Alzheimer disease" at first mention


Supporting Information:

The structures of the compounds in SMILES format are not half as useful as 3D-structures would have been. A link to the precise query used to download those 3D-structures from the http://tcm.cmu.edu.tw/ would probably be more useful. I do not know if the TCM database allows re-distribution of their data: if they do, I would advise you to include the 3D-structures of the 1654 molecules instead of their SMILES strings. If such redistribution is not allowed, I think that providing the SMILES strings of the molecules in each cluster in separate files would be much more useful than the current massive SMILES file.

The Supporting information includes some data on clinical reports of the efficay of traditional Chinese Medicine prescriptions for Alzheimer Disease. The most important data (titles of the studies, their references/sources and results) are unfortunately missing, which severely hampers the usefulness of that information.

Reviewer 2 ·

Basic reporting

no comments

Experimental design

There aren't sufficient informations to replicate

Validity of the findings

no comments

Additional comments

in my opinion, the authors should better describe AD, justifying the choice of targets. They also lack information on how they were obtained.

Reviewer 3 ·

Basic reporting

no comment

Experimental design

no comment

Validity of the findings

no comment

Additional comments

The paper is clearly written, well organized and should be accepted for publication after minor revision.
The detailed validation procedure of the docking results should be presented. The docking results should be visualized, at least in the Supplementary material. Also, it would be useful if the lipophilicity of the selected compounds is presented (logP).

Reviewer 4 ·

Basic reporting

see attached annotated PDF

Experimental design

see attached annotated PDF

Validity of the findings

see attached annotated PDF

Additional comments

see attached annotated PDF

Annotated reviews are not available for download in order to protect the identity of reviewers who chose to remain anonymous.

---

## Round 0.2 · Major Revisions

After the revision again we have contrasting opinions from the reviewers - from accept to reject. We have to keep high quality of the journal and take into account all the remarks. Thus, the manuscript needs some revision (minor or major revision).

Minus of the work is lack of experimental evidence of the possible drug effects.

Good suggestion is to make Supplement / Supporting information file. Please write the conclusion of the manuscript suggesting your strategy to find anti-AD drugs as a method of hypothesis. Thus, the work could be stated as some "Method" or theoretical paper.

The problem of such drug search is too complex to be solved. Please discuss possible limits of the suggested approach in the paper - does it possible scan the suggested drugs? How many variants of such drugs exist based on known natural compounds?

The conclusions should be clear to wider reader audience.

·

Basic reporting

The new version of this manuscript is much improved. I think, however, that the presentation of results and conclusions still needs some more work to ensure that your research is appropriately communicated.

Experimental design

line 285: I am not completely convinced that dividing the full 1476 molecules in clusters is very informative. Wouldn't it be more intuitive to search for clusters in the sets of ligands that bind to each individual protein (i.e. to see if there were very different scaffolds able to bind SIRT1, lyn,....,etc.) ?

Validity of the findings

lines 204-207: Authors state" The docking results of these compounds indicated that almost all of the docking energies of the top 0.5% of compounds bound to target proteins were superior to those of their embedded ligands (Fig. 1). Thus, the docking results are reliable, and the 1,476 compounds can be considered candidate compounds for AD." I am afraid that the second sentence does not necessarily follow from the first one, and that at most you could say " [...]. Assuming the docking results are reliable, these 1,476 compounds can be considered candidate compounds for AD". To show that your docking results are reliable, you would have to show that experimentally-confirmed weak-binding analogues of the original binding ligands do not have better docking energies than the original ligands. Such benchmarking has probably been performed by the original developers of the docking algorithm used, either in the original publication of the method or in closely related work. I think you should refer to the false-positive rates (or receiver operating characteristics) described in those works for an estimate of the likely percentage of true positives among the 1476 compounds.

lines 222 ff.: Authors compare docking energies of their candidates to those of four AD drugs. They do not, however, state whether those drugs bind to any of the tested proteins, and the relevance of these comparisons is therefore not clear.

Additional comments

lines 210-217: I do not think that listing the residues in the binding pocket of a single protein is very useful: an image with a side-by-side-comparison of best TCM ligand and original ligand would probably be easier to understand (actually, I think that providing the PDB coordinates of each protein with embedded original ligand and best TCM ligands as Supporting information would be ideal). Anyway, you should report the range of similarities you find (Tc betwwen 0.06 and 0.48) instead of the single Tc for one protein.

lines 218-221 . State the reason behind the choice of only SIRT1 for pharmacophore detection

lines 262: authors refer to "existing drugs". Do they mean "existing anti-AD drugs" or "existing molecules present in at least one of the world's pharmacopoeias"? Please clarify.

Reviewer 2 ·

Basic reporting

In my opinion, the manuscriptt can be published.

Experimental design

In my opinion, the manuscriptt can be published.

Validity of the findings

In my opinion, the manuscriptt can be published.

Additional comments

In my opinion, the manuscriptt can be published.

Reviewer 3 ·

Basic reporting

The authors corrected the manuscript according to the suggestions.
The paper should be accepted for publication.

Experimental design

The authors corrected the manuscript according to the suggestions.
The paper should be accepted for publication.

Validity of the findings

The authors corrected the manuscript according to the suggestions.
The paper should be accepted for publication.

Additional comments

The authors corrected the manuscript according to the suggestions.
The paper should be accepted for publication.

Reviewer 4 ·

Basic reporting

Authors have done some improvement, mostly on the editorial and formatting side.

Experimental design

I am still concerned with the experimental design of this work. authors have used a large number of TCM compounds to dock against large number of targets - it is like you comparing apples with tomatoes. That is why it is hard to draw out a concise conclusion or outcome of this work. I see that authors did not provide a proper conclusion with the key findings/outcome. I find it hard to come out with a message from this work that can help medicinal chemists or biologists do further research or expand on the outcome/results created here. Authors stated in the discussion/results that "Most of these
compounds are abundantly found in plants used for treating AD in China" - so where is the novelty here. the outcome of the study was found to be exactly what is actually known for these compounds and what they have been used for as anti AD!! the study did not provide any additional information, no additional structural or molecular description of the identified compounds, no precise description of the binding mode/mechanisms of the identified compounds.I am sorry, yes authors have done quite a bit of calculations with lots of numbers, however, these numbers don't provide any useful information to the scientific community.

Validity of the findings

Please see above comments

Additional comments

please see above

---

## Round 0.3 · accepted · Accept

This manuscript had a good revision. One reviewer had critical remarks that I believe are not related to the quality of the work, and so I am happy to Accept the manuscript with this revision. Thank you for patience. I believe the work could be published in the present form.

# ·

Basic reporting

see below

Experimental design

see below

Validity of the findings

see below

Additional comments

All my objections have been appropriately answered. I think the manuscript can be published as-is.